# Optimization of Enzyme-Assisted Mechanical Extraction Process of *Hodgsonia heteroclita* Oilseeds and Physical, Chemical, and Nutritional Properties of the Oils

**DOI:** 10.3390/foods12020292

**Published:** 2023-01-08

**Authors:** Jirachaya Piseskul, Uthaiwan Suttisansanee, Chaowanee Chupeerach, Chanakan Khemthong, Sirinapa Thangsiri, Piya Temviriyanukul, Yuraporn Sahasakul, Chalat Santivarangkna, Rungrat Chamchan, Amornrat Aursalung, Nattira On–nom

**Affiliations:** Food and Nutrition Academic and Research Cluster, Institute of Nutrition, Mahidol University, Salaya, Phuttamonthon, Nakhon Pathom 73170, Thailand

**Keywords:** pretreatment, seed oil, response surface methodology, fatty acid profile, neglected and underutilized crops, sustainability

## Abstract

*Hodgsonia heteroclita* subsp. *Indochinensis* W.J.de Wlide & Duyfjes (or Making in Thai) is a neglected and underutilized crop (NUC) with high fat containing nuts. In this study, the enzyme-assisted mechanical extraction of *H. heteroclita* seed oil was investigated using response surface methodology (RSM) to predict the optimal fat extraction conditions. The most efficient enzyme used in the experiment was a mixture of Flavourzyme^®^ and Viscozyme^®^ (1:1, *w*/*w*). The predicted maximum oil yield was 46.44%, using the following extraction conditions: 2.98% (*w*/*w*) enzyme loading, 48 °C incubation temperature and 76 min of incubation time. *H. heteroclita* seed oil obtained from heat and enzymatic pretreatments exhibited the highest lightness and viscosity. The chemical properties of this seed oil, including water and volatile compounds (≤0.2% *w*/*w*), acid value (≤4.0 mg KOH/g), peroxide value (≤15 mEq of active oxygen/kg) and soap content (≤0.005% *w*/*w*), were within the acceptable levels specified by the Codex Alimentarius (2019). *H. heteroclita* seed oil obtained from heat and enzymatic pretreatments contained the highest content of δ-tocopherol (88.29 mg/100 g) and omega-6 fatty acids (48.19 g/100 g). This study is the first to report on the enzyme-assisted mechanical extraction of *H. heteroclita* oilseeds as a promising plant material for vegetable oil production.

## 1. Introduction

Nowadays, many vegetable oils such as palm, soya bean, rapeseed, sunflower, cottonseed and olive are commercially available [1]. The report from IndustryARC^TM^ predicted that the market size of global vegetable oil will reach USD 3700 million by 2026 [2]. However, the growing demand for vegetable oils requires agricultural intensification and increased global natural land clearing to improve production yield, leading to negative effects on sustainability. Thus, using neglected and underutilized crops (NUCs) to replace staple vegetable oils is currently of interest and can help to enhance sustainability by avoiding the uniform cropping system and adding more value to NUCs, consequently helping to generate income for local people [3].

*Hodgsonia heteroclita* subsp. *Indochinensis*, family Cucurbitaceae, is a perennial climber plant that grows well in the hilly terrains of southern Asia, such as in Bangladesh, Bhutan, Cambodia, Laos, Myanmar, Thailand, Vietnam and India [4]. In Thailand, *H. heteroclite*, or Making in Thai, is classified as an NUC and listed as an endangered species by the Plant Genetic Conservation Project under the initiative of Her Royal Highness Princess Maha Chakri Sirindhorn (RSPG). This wild plant is only found in Chiang Mai, Chiang Rai, Lampang, Phrae and Nan Provinces in Thailand, at the elevation of 400–1500 m above sea level. The edible seed contains high amounts of fat (32.5–33.5%), protein (26.7–27.6%), carbohydrate (19.1–23.0%), minerals, vitamin E and phenolic compounds [5]. Currently, its presence and consumption are only recognized by locals, and the optimized conditions for the effective extraction of seed oil remained uninvestigated. Therefore, the industrial application of *H. heteroclita* for the production of vegetable oil can increase its value while creating sustainability.

Commonly, organic solvent extractions are methods for oil extraction in the food industry. These provide high extraction efficiency, but require long extraction time with high investment and operation cost. Chemical usage can also cause environmental issues. Another well-known conventional method is mechanical extraction, by applying pressure to press the oil out of seeds, as an easy operation with simple maintenance. This method is preferable due to the higher remaining amounts of some beneficial components in the extracted oil, such as essential fatty acid, vitamins, carotenoids, glycolipids and sterols. However, a lower oil yield is produced compared to solvent extraction [6]. To enhance the extraction yield, thermal and enzymatic pretreatments are generally introduced to the extraction process. Thermal pretreatment softens plant materials, resulting in easier oil extraction. However, the application of high temperature can also reduce nutritional value and oil quality [7,8]. To preserve valuable components, enzymatic pretreatment is employed due to its mild conditions. The hydrolysis activity of enzymes plays an important role in disrupting the cell wall structure of the plant materials, leading to easier oil extraction. Previous studies reported that papain (protease from papaya latex) significantly increased peanut and sacha inchi seed oil yields, while the addition of various carbohydrases (such as cellulase, pectinase and β-glucanase) significantly improved efficiency in cottonseed and sunflower seed oil extractions [9,10,11,12]. Previous studies also reported that a combination of heat and enzymatic pretreatments in the pressing process successfully extracted oil from rose-hip, Chilean hazelnut and borage, resulting in higher oil yields (72–98%) [13,14,15,16].

Based on previous information, this study optimized enzyme-assisted mechanical extraction from *H. heteroclita* oil seeds to obtain the highest oil extraction yield with desirable oil quality using response surface methodology (RSM). This method is generally used to study the influence of multiple factors and their interactions on response variables, and can identify the optimal operating conditions for maximizing yield and quality. RSM is a more effective method compared to classical factorial analysis, since it requires fewer experiments to study the effects of all factors and their combinations, leading to reduced experimental time and cost [17]. In this study, the independent factors including enzyme loading, incubation temperature and incubation time, were varied to achieve optimized oil yield. The physical, chemical and nutritional properties of *H. heteroclita* seed oil obtained from the optimized enzyme-assisted extraction conditions (heat and enzymatic pretreatments) were determined and compared with no pretreatment and heat-only pretreated seed oils. Information gained from this study can be used for the extraction of *H. heteroclita* seed oil at the industrial level. Applications of *H. heteroclita* seed oil in real food products such as salad dressing, frying and spreads could also be examined in further study.

## 2. Materials and Methods

### 2.1. Sample Collection and Preparation

*H. heteroclita* fruits (Figure 1a) were collected from forests near Hui-Nam-Guen village, Mae-Jedi-Mai sub-district, Weing-Pa-Pao district, and Chiang Rai Province, Thailand, between August and October, 2021. The samples were authenticated and identified as *Hodgsonia heteroclita* subsp. *Indochinensis* W.J.de Wlide & Duyfjes by Assist. Prof. Dr. Bhanubong Bongcheewin, Faculty of Pharmacy, Mahidol University (Bangkok, Thailand), according to a reliable reference [18]. The plants were deposited at Sireeruckhachati Nature Learning Park, Mahidol University, Nakhon Pathom, Thailand for voucher specimens (PBM 005646). *H. heteroclita* oil seeds (Figure 1b,c) were prepared by dehulling the hard shell and removing the seed coat. To control the quality of seeds, nutritive values were determined using standard protocols of the Association of Official Analytical Chemists (AOAC), as described in Section 2.4.3. The seeds contained 18.77 ± 0.30% moisture content, 2.41 ± 0.31% ash, 13.79 ± 0.61% total carbohydrates (comprising 13.42 ± 0.46% dietary fiber), 26.58 ± 0.64% protein and 39.16 ± 0.27% total fat. Total calories and calories from fat were 513.90 ± 2.59 kcal/100 g and 352.44 ± 2.42 kcal/100 g, respectively. The samples were stored at −20 °C for further extraction.

### 2.2. Effect of Different Enzymes on H. heteroclita Seed Oil Extraction

The seeds were cut into 0.5 × 0.5 cm^2^ and crushed using a Philips Daily Collection Blender (450 W, 0.7 bowl) (model HR1393, Philips Electronics (Thailand) Co., Ltd., Bangkok, Thailand) for 30 s per 100 g sample. The crushed seeds (pH ~5.7) were then pressed into an HDPE plastic bag and dried using a Binder GMBH hot air oven (model FFD115/E2, Binder GmbH, Tuttlingen, Germany) at 55 °C for 90 min to obtain approximately 10% moisture content (heat pretreatment). Then, enzymatic pretreatment was performed according to the method modified from Sharma et al. (2015) [19], using commercially available enzymes including Flavourzyme^®^ (available strength 500–1000 Leucine Amino Peptidase Units/g) and Viscozyme^®^ (available strength 100 Fungal Beta Glucanase Units/g), which were provided by Novozymes Company (Bagsværd, Denmark). Flavourzyme^®^ is typically active at pH 5–7 and temperature 50 °C, while Viscozyme^®^ is typically active at pH 3.5–5.5 and temperature 25–55 °C [20,21]. The dried sample was incubated with 1% (*w*/*w*) of either Flavourzyme^®^, Viscozyme^®^ or a mixture of Flavourzyme^®^ and Viscozyme^®^ at a ratio of 1:1 (*w*/*w*) at 40 °C for 1 h. Enzymatic activity was then immediately inactivated using a hot air oven at 90 °C for 5 min. After that, seed oil was extracted using a rotary screw extractor (model T3, Nature Health and Innovation Co., Ltd., Saraburi, Thailand) at 50–60 °C. The extracted mixture was subjected to centrifugation using a Hettich^®^ ROTINA 38/38R centrifuge (Andreas Hettich GmbH & Co., KG, Tuttlingen, Germany) at 2810× *g*, 25 °C for 15 min. The oil layer (upper layer) was collected and filtered through Whatman^TM^ grade 91 filter paper (10 µm).

Mechanical extraction without heat and enzymatic pretreatment and solvent extraction by a Soxtec method were also performed as the controls. Mechanical extraction without heat and enzymatic pretreatment was performed as described above without heat pretreatment and addition of enzyme, while the Soxtec extraction was conducted by Central Laboratory (Thailand) Co., Ltd. (Bangkok, Thailand) according to AOAC (2019) [22] 922.06. Oil yield and extraction efficiency were calculated using Equations (1) and (2), respectively.
(1)$$\mathrm{Oil}\;\mathrm{yield}\;\left(\%\right)=\frac{\mathrm{Weight}\;\mathrm{of}\;\mathrm{extracted}\;\mathrm{oil}\;(g)}{\mathrm{Weight}\;\mathrm{of}\;\mathrm{seed}\;\mathrm{used}\;(g)}\times 100$$
(2)$$\mathrm{Extraction}\;\mathrm{efficiency}\;\left(\%\right)=\frac{\mathrm{Oil}\;\mathrm{yield}\;\mathrm{obtained}\;\mathrm{by}\;\mathrm{enzyme}-\mathrm{assisted}\;\mathrm{extraction}}{\mathrm{Oil}\;\mathrm{yield}\;\mathrm{obtained}\;\mathrm{by}\;\mathrm{Soxtec}\;\mathrm{extraction}}\times 100$$

### 2.3. Optimization of Enzyme-Assisted Extraction Conditions Using Response Surface Methodology (RSM)

Enzyme-assisted extraction of *H. heteroclita* seed oil was optimized utilizing RSM with a Box–Behnken design (Design-Expert software version 13, Stat-Ease, Inc., Minneapolis, MN, USA) following Nguyen et al. (2020) [11] with some modifications. The enzymatic pretreatment was conducted in a hot air oven using different enzyme loading (1–5% (*w*/*w*) based on sample weight), incubation temperature (30–50 °C) and incubation time (0.5–1.5 h). The oil yield was calculated using Equation (1). The relationship between extraction conditions (independent variables, X) and oil yield (response, Y) was modeled using Equation (3), as follows:(3)$$Y\;=a_{0}+\sum_{i=1}^{3}a_{i}X_{i}+\sum_{i=1}^{3}a_{\mathrm{ii}}X_{i}^{2}+\sum_{i=1}^{2}\sum_{j=2}^{3}a_{\mathrm{ij}}X_{i}X_{j}$$
where Y is the oil yield (%) and X_1_, X_2_ and X_3_ are enzyme loading, incubation temperature and incubation time, respectively, and a_0_, a_i_, a_ii_ and a_ij_ are the constant, linear, quadratic and interaction coefficients, respectively. Design-Expert software (version 13) was used to develop the RSM model and perform analysis of variance and the goodness-of-fit test (coefficient of determination (R^2^) in regression analysis) between the predicted and experimental oil yields. The proposed model was used to optimize extraction conditions to achieve the highest oil yield. The optimized extraction conditions were validated by the one-sample *t*-test using IBM SPSS Statistics 26.

### 2.4. Determination of Physical, Chemical and Nutritional Properties of H. heteroclita Seed Oils

Seed oils obtained from mechanical extraction with (i) heat and enzymatic pretreatments (optimized condition), (ii) heat pretreatment (without enzymatic reaction) and (iii) no pretreatment (without heat and enzymatic pretreatments) were compared in terms of physical, chemical and nutritional properties as follows.

#### 2.4.1. Physical Properties

The color of seed oils was analyzed using a ColorFlex^®^ EZ Spectrophotometer (Hunter Associates Laboratories, Inc., Reston, VI, USA) with a light source of D65/10° as according to Keseke et al., (2021) with some modification [23]. The lightness (L*), redness (a*) and yellowness (b*) values were measured, and the yellowness index (YI) was calculated using Equation (4).
(4)$$\mathrm{YI}\;=\frac{142.86b*}{L*}$$

Viscosity of the seed oils was also measured using an RVT-DVII Brookfield Digital Viscometer (Brookfield Engineering Laboratories, Inc., Stoughton, MA, USA). Apparent specific gravity was analyzed according to AOAC (2019) 990.212, while refractive index was determined following AOAC (2019) 921.08. These experiments were performed by the Analytical Service at the Department of Science Service (Bangkok, Thailand).

#### 2.4.2. Chemical Properties

Moisture and volatile compounds were determined by the drying method following ISO 662: 2016 [24]. Acid and peroxide values were analyzed by the titration technique according to an in-house method based on AOAC (2019) 940.28 and 965.33, respectively. Similarly, iodine value was determined by the Wijs method using a titration technique following AOAC (2019) 993.20. Saponification value, as the amount of alkali used to saponify a specific amount of sample, was determined according to AOCS [25] (2009) Cd 3–25. By contrast, unsaponifiable matters, including dissolvable substances in fats and oils that cannot be saponified by the usual caustic treatment (e.g., higher aliphatic alcohols, sterols, pigments and hydrocarbons), were analyzed according to AOCS (2009) Ca 6b-53. Soap content was determined using a titration technique following AOCS (2009) Cc 17–95. Contaminants including iron (Fe), copper (Cu) and lead (Pb) were analyzed using graphite furnace atomic absorption spectroscopy (GFAAS) following an in-house method based on AOAC (2019) 999.11. Arsenic (As) was determined using atomic absorption spectroscopy (AAS) and anodic stripping voltammetry (ASV) according to an in-house method based on AOAC (2019) 986.15. Aflatoxin B1 was analyzed using thin-layer chromatography-fluorodensitometry and silica gel chromatography following an in-house method based on AOAC (2019) 993.17. The limit of detection (LOD) of Fe was 0.20 mg/kg, Cu was 0.04 mg/kg, PB was 0.03 mg/kg, As was 0.001 mg/kg and Aflatoxin B1 was 0.4 µg/kg. All chemical properties were performed by the Analytical Service at the Department of Science Service (Bangkok, Thailand).

#### 2.4.3. Nutritional Properties

Nutritional properties of the seed oils were determined for moisture content, fat, protein, carbohydrate and ash. Moisture content was determined using a hot air oven following AOCS Ca 2C-25 (1997). Ash was analyzed using a Carbolite CWF 1100 muffle furnace (Carbolite Gero Ltd., Hope, UK) according to AOAC (2019) 920.153. Fat was determined using acid hydrolysis following AOAC (2019) 922.06. Carbohydrates and protein were analyzed according to an in-house method TE-CH-169 based on the Method of Analysis for Nutrition Labeling (1993) P.108 and AOAC (2019) 981.10, respectively. The fatty acid profile was determined using gas chromatography according to an in-house method TE-CH-208 based on AOAC (2019) 996.06. Lastly, vitamin E was analyzed by high-performance liquid chromatography (HPLC) following Azrina et al., (2008) [26] and Chen and Bergman (2005) [27]. All nutritive values were conducted by the Analytical Service at the Central Laboratory (Thailand) Co., Ltd., Bangkok, Thailand.

### 2.5. Statistical Analysis

The results obtained from the effect of different enzymes on seed oil extraction and determination of physical, chemical and nutritional properties of *H. heteroclita* seed oils were calculated as mean ± standard deviation (SD) of triplicated experiments on three individual sets of samples (*n* = 3), and compared by one-way analysis of variance (ANOVA) with Duncan’s post hoc test at *p* < 0.05 using IBM SPSS Statistics 26 software (IBM Corp., Armonk, NY, USA).

## 3. Results

### 3.1. Effect of Different Enzymes on H. heteroclita Seed Oil Extraction

The oil yield (%) and extraction efficiency (%) of *H. heteroclita* seed oil were obtained from mechanical extraction with and without enzymatic pretreatment and compared to Soxtec extraction, as shown in Table 1. Oil yield ranged between 33.64 and 39.16%. Soxtec extraction provided the highest oil yield (up to 1.6-fold higher than the others). However, no significant differences in oil yield were observed among Soxtec extraction and mechanical extraction with enzymatic pretreatment of Flavourzyme^®^ alone or the combination of Flavourzyme^®^ and Viscozyme^®^ (1:1, *w*/*w*). Oil yield of mechanical extraction with enzymatic pretreatment of Flavourzyme^®^ was slightly higher than using Viscozyme^®^ but insignificantly different. Mechanical extraction without enzymatic pretreatment exhibited the lowest oil yield. Extraction efficiency (%) was calculated as the average value of oil yield obtained from the different extraction methods compared to Soxtec extraction (100% extraction efficiency). Results of extraction efficiency showed the same trends with oil yield. Mechanical extraction with enzymatic pretreatment of Flavourzyme^®^ and Viscozyme^®^ (1:1, *w*/*w*) mixture gave the highest extraction efficiency (99.44%). Thus, the combination of Flavourzyme^®^ and Viscozyme^®^ (1:1, *w*/*w*) was selected for further study using RSM.

### 3.2. Optimization of Enzyme-Assisted Extraction Conditions Using Response Surface Methodology (RSM)

#### 3.2.1. RSM Model Development

A Box–Behnken design was applied to develop the RSM model, using three factors (independent variables) each with three levels (−1 to 1). The independent variables were enzyme loading (X_1_), incubation temperature (X_2_) and incubation time (X_3_). The dependent variable or response was oil extraction yield (Y). Table 2 shows the matrix design and experimental data of the obtained oil yields. A low coefficient of variance (2.45%) of the central experiments (run nos. 13–15) represented consistent experimental performance.

The analysis of variance (ANOVA) of the regression model is presented in Table 3. The model was statistically significant (*p* = 0.0038), while *p*-value of the lack of fit was higher than 0.05. The coefficient of determination (R^2^) was 0.96, indicating a strong relationship between the predicted and experimental responses, while the difference between adjusted R^2^ and predicted R^2^ was less than 0.2. Adequate precision was recorded, while estimation of the signal-to-noise ratio was greater than 4 (13.329), signifying that the signal was adequate to navigate this established model. High reproducibility of the model was assured by a small coefficient of variance (CV < 10%) of the center points.

The model (Table 3) showed that the percentage of oil yield is significantly (*p* < 0.05) affected by the linear terms of enzyme loading (x_1_) and incubation time (x_3_), as well as by the quadratic terms of enzyme loading (x_1_^2^) and incubation time (x_3_^2^). The quadratic relationship between input variables and experimental responses was generated to model with a 0.05 level of significance, as follows Equation (5).
Y = 10.86906 + 8.97250X_1_ + 0.355306X_3_ − 1.06073X_1_^2^ − 0.001989X_3_^2^(5)

#### 3.2.2. Effect of Extraction Conditions on Oil Yield

The effects of enzyme loading and incubation temperature on oil extraction yield are shown in Figure 2(A1,A2), with a fixed period of incubation time at 60 min. At low enzyme loading (less than 3%), oil extraction yield showed a decreased trend. After the highest oil extraction yield was reached with a utilization of approximately 3% enzyme loading, yield continued decreasing even with higher enzyme loading. By contrast, incubation temperature did not strongly impact oil extraction yield, corresponding to the statistical data (ANOVA) as the *p*-values of incubation temperature were insignificant at *p* ≥ 0.05 (*p* = 0.2280). Figure 2(B1,B2) indicated the influence of enzyme loading and incubation time on oil extraction yield, while incubation temperature was maintained at the center point (40 °C). A similar trend was observed for enzyme loading and incubation temperature, with optimized oil extraction yield achieved at 3% enzyme loading. However, in this case, the oil extraction yield was also affected by incubation time. Longer incubation time led to higher oil extraction yield, which reached a plateau when incubation time was longer than 80 min. The combined effect of incubation temperature and incubation time on oil extraction yield is shown in Figure 2(C1,C2), with enzyme loading controlled at a constant value (3%). Oil extraction yield was only minimally affected by incubation temperature, corresponding to the results shown in Figure 2(A1,A2). Incubation time exhibited a significantly positive effect on oil extraction yield at *p* = 0.0018, while higher oil extraction yield was predicted at a longer incubation time, as shown in Figure 2(B1,B2).

#### 3.2.3. Determination of Optimal Conditions

Optimal conditions were predicted to maximize oil extraction yield by the RSM model, using Equation (5). The highest oil extraction yield was predicted as 46.44%, using 2.98% (*w*/*w*) enzyme loading, 48 °C incubation temperature and 76 min incubation time.

To validate the extraction conditions predicted by RSM, actual experiments were performed under the optimal extraction conditions stated above. The actual experiments gave an oil extraction yield of 42.22 ± 2.73%, insignificantly different compared to the predicted value (*p* = 0.116) by the regression model. Therefore, the predicted model (Equation (5)) was practicable and accurate to predict the oil extraction yield from *H. heteroclita* oil seed. Moreover, oil yield from the optimal extraction conditions was higher than under Soxtec extraction (39.16 ± 0.27%).

### 3.3. Physical, Chemical and Nutritional Properties of H. heteroclita Seed Oils

In the preliminary experiment, mechanical extraction with enzymatic pretreatment (without heat pretreatment) showed that no oil yield was obtained due to the high moisture content (>20%). Therefore, seed oils obtained from mechanical extraction with (i) heat and enzymatic pretreatments (optimized conditions), (ii) heat pretreatment (without enzymatic reaction) and (iii) no pretreatment (without heat and enzymatic pretreatments) were compared in terms of physical, chemical and nutritional properties.

#### 3.3.1. Physical Properties

Physical properties of *H. heteroclita* seed oils, including color, viscosity, specific gravity and refractive index, are shown in Table 4. *H. heteroclita* seed oil obtained from heat and enzymatic pretreatments exhibited slightly higher lightness (L*) than the others, while redness (a*) was the highest (1.4–1.6-fold higher than the others). The lowest L* and a* values were reported in *H. heteroclita* seed oils obtained from heat pretreatment and no pretreatment, respectively. For yellowness (b*) and yellowness index (YI), *H. heteroclita* seed oil obtained from heat pretreatment presented the highest b* value and YI (1.4–1.5-fold higher than the others), while the lowest b* values and YI were reported in *H. heteroclita* seed oil obtained from heat and enzymatic pretreatments.

The highest viscosity was observed in seed oil extracted with heat and enzymatic pretreatments (1.4–1.6-fold higher than the others), while the lowest viscosity was in only heat pretreatment. No significant differences in specific gravity (0.92) and refractive index (1.47) were presented among seed oils extracted under different extraction conditions.

#### 3.3.2. Chemical Properties

Chemical properties of *H. heteroclita* seed oils obtained from different mechanical extraction pretreatment conditions are shown in Table 5. Water and volatile compounds ranging 0.06–0.09% (*w*/*w*) were insignificantly different among all seed oils extracted under different pretreatments. Similarly, insignificantly different iodine contents were observed (105.75–106.40 g I_2_/100 g). However, acid, peroxide, saponification, unsaponifiable matters and soap contents of all seed oils extracted under different pretreatments varied significantly. Seed oil extracted without pretreatment exhibited significantly higher acid (2.1–3.7-fold higher than the others) and peroxide values (4.2–5.8-fold higher than the others). For saponification, slightly but significantly higher values were recorded in seed oils extracted without heat and with heat pretreatments than those that underwent heat and enzymatic pretreatments. Similar results were observed in soap content. Seed oils extracted without heat and with heat pretreatments exhibited significantly higher soap contents (6.3–8.8-fold higher) than those that underwent heat and enzymatic pretreatments. By contrast, the unsaponifiable matters of seed oils that underwent heat and enzymatic pretreatments were significantly higher (2.2–3-fold) than those that underwent heat pretreatment and no pretreatment, respectively. Contaminants including iron (Fe), copper (Cu), lead (Pb), arsenic (As) and aflatoxin B1 were not found in all the seed oils.

#### 3.3.3. Nutritional Properties

Proximate analyses of *H. heteroclita* seed oils obtained from different mechanical extraction pretreatment conditions are shown in Table 6. Fat contents in all seed oils were equal at 100 g/100 g, while 0.01 g/100 g of moisture, ash and carbohydrate as well as <0.10 g/100 g of protein were detected, indicating a high purity of all extracted oils. Vitamin E analysis indicated that each seed oil contained different amounts and forms of tocopherol and tocotrienol due to different extraction conditions. δ-Tocotrienol was the dominant form found in seed oils extracted using no pretreatment and heat pretreatment, with trace amounts of β-tocopherol and δ-tocopherol. Seed oils extracted using no pretreatment also contained trace amounts of α-tocopherol and β-tocotrienol, while those extracted with heat pretreatment also possessed trace amounts of γ-tocopherol. δ-Tocopherol was present in all seed oils, and as the predominant vitamin E in seed oils extracted with heat and enzymatic pretreatments. Moreover, small amounts of γ-tocopherol, α-tocotrienol, β-tocotrienol and γ-tocotrienol were also present in seed oil extracted with heat and enzymatic pretreatments, with contents higher than those extracted with no pretreatment and with heat-only pretreatment.

Fatty acid compositions and omega-3,6,9 fatty acids of *H. heteroclita* seed oils from different mechanical extraction pretreatment conditions were also analyzed, as shown in Table 7. Results revealed that the fatty acid composition of all seed oils was similar, comprising 60% unsaturated fat, separated into monounsaturated fatty acids (10%, mainly oleic acid (C18:1)) and polyunsaturated fatty acids (50%, mainly linoleic acid (C18:2)). Another 34% of fatty acid composition was saturated fat, including 27% palmitic acid (C16:0) and 7% stearic acid (C18:0). For unsaturated fat, seed oil extracted with heat pretreatment contained slightly but significantly higher (1.1-fold) monounsaturated fatty acid contents than the others, while polyunsaturated fatty acid contents were highest in seed oil extracted with heat and enzymatic pretreatments. Insignificantly different oleic acid (C18:1) and linoleic acid (C18:2) contents were observed in all seed oils. No significant differences in saturated fat contents among seed oils extracted using different pretreatment techniques were observed; however, stearic acid (C18:0) in seed oil under heat pretreatment was significantly higher (1.1-fold) than in the others.

All seed oils contained high amounts of omega-6 fatty acid, corresponding to high linoleic acid (C18:2) content, followed by omega-9 fatty acid, which is related to oleic acid (C18:1) content. Insignificantly different oleic acid (C18:2) contents were observed among seed oils extracted using different pretreatment techniques, but omega-9 fatty acids in seed oil extracted with heat pretreatment were statistically higher (1.1-fold) than the others. Omega-3 fatty acid was found in the least amounts, in accordance with a small quantity of α-linolenic acid (ALA) (C18:3). Similarly, omega-3 fatty acid contents in seed oil under no pretreatment and the combination of heat and enzymatic pretreatments were slightly but significantly higher than those that underwent only heat pretreatment.

## 4. Discussion

*H. heteroclita* is a wild plant with potential application as a plant material in vegetable oil production. It contains high fat content (>30%) with high unsaturated fatty acids [5]. Mechanical extraction using a rotary screw extractor is often used to preserve nutrient contents, and this technique is also eco-friendly. However, lower oil extraction yield is normally achieved compared to solvent extraction. Therefore, pretreatment methods are introduced into mechanical extraction to increase oil yield. Enzymatic pretreatment aids the hydrolysis of plant cell walls and the degradation of oil seed membrane structure, resulting in improved structure permeability and increasing oil release during mechanical extraction [28,29]. Oil extraction from *H. heteroclita* oilseeds has remained largely uninvestigated. This is the first report focusing on developing heat and enzyme-assisted mechanical methods to optimize extraction yield from *H. heteroclita* oilseeds using RSM, while maintaining oil quality.

The commercially available enzymes Flavourzyme^®^ and Viscozyme^®^ were selected based on their widely reported applications. More than a 70% increase in oil yield was reported in Kalahari melon seed and rapeseed oil extraction using Flavourzyme^®^-assisted aqueous extraction [30,31]. Viscozyme^®^ also enhanced extraction yield of *Irvingia gabonensis* seed kernels oil and argan oil extraction yield at up to 70% using enzyme-assisted aqueous extraction [32,33]. Therefore, higher yield is detected in oil extraction with enzymatic pretreatment. Previous results found that higher volumes of canola, rosehip and hazelnut oil were obtained after application of enzymatic hydrolysis [13,16,34]. Among different enzymatic pretreatments, the mixture of Flavourzyme^®^ and Viscozyme^®^ (1:1, *w*/*w*) provided the highest oil yield (38.94%), due to their hydrolytic activity on plant cell walls and the membrane structure of oil bodies [11] as the seed is composed of 26.58% protein and 13.79% carbohydrate. Therefore, this enzyme mixture was chosen for further experiments to optimize oil yield under different mechanical extraction conditions using RSM.

A Box–Behnken RSM model was chosen because this technique reduces the number of experiments and only requires three levels (−1 to 1) and three factors (independent variables), leading to lower costs and reduced labor [17,35]. Thus, a suitable temperature for oil extraction should be in the range of the recommended temperature for the employed enzymes, i.e., 50 °C for Flavourzyme^®^ and 25–55 °C for Viscozyme^®^ [20,21]. Longer incubation time had a positive effect on oil extraction yield, related to enzyme activity during the incubation phase. Our results indicated that when oil seeds were incubated for longer than 80 min, oil extraction yield reached a plateau. This incidence was related to the release of proteins, glycolipids and phospholipids as a result of the enzymatic activity, leading to the formation of protein-lipid aggregates and the stabilization of an emulsion or the limiting substrate [11,36]. Similar results were recorded in oil extraction from *Pinus pumila* seed kernels, perilla seeds and sacha inchi seeds. When the optimal oil extraction yield was reached under a certain incubation time, this remained unchanged or slightly decreased, even at longer incubation [11,37,38]. Based on the RSM model and Equation (5), maximum oil extraction yield was predicted as 42.22% under enzyme-assisted extraction conditions of 2.98% enzyme loading, 48 °C incubation temperature and 76 min incubation time. A *p*-value of < 0.05 indicated the statistical significance of the established model, while the coefficient of determination (R^2^ = 0.96) signified an excellent correlation between predicted and experimental values. Therefore, the developed model provided the optimal extraction conditions to obtain the highest oil yield.

*H. heteroclita* seed oil obtained from the optimal enzyme-assisted mechanical extraction (heat with enzymatic pretreated mechanical extraction) conditions was subsequently analyzed for physical, chemical and nutritional properties, together with *H. heteroclita* seed oils obtained from mechanical extraction with no pretreatment, and heat-only pretreatment. For physical properties, seed oil extracted using heat pretreatment exhibited the highest b* value. An increase in b* value was also observed in other oil extractions utilizing different heat pretreatments. The color change was related to Maillard reactions or phospholipid degradation [39,40]. Seed oil viscosity and ability to flow at certain temperatures significantly decreased after heat pretreatment because of the degradation of oil molecules after conventional heating [41]. Viscosity increased with the molecular weight of fatty acids, but decreased with increasing unsaturation and temperature. In our study, the lowest viscosity was observed in *H. heteroclita* seed oils extracted using heat pretreatment, due to reduced intermolecular attraction between the long-chain structures of triacylglycerol molecules [42]. The viscosity of *H. heteroclita* seed oils extracted using heat pretreatment and without treatment (54.91 and 62.71 cps) were in the range of palm olein oil, coconut oil and soybean oil (48.1–69.2 cps), while the viscosity of *H. heteroclita* seed oil extracted using heat and enzymatic pretreatments (88.42 cps) was similar to olive oil (84 cps) [43]. Our results suggested insignificantly different specific gravity among seed oils extracted under different pretreatments. Specific gravity is a ratio of oil weight to reference substance (water) weight at an equal volume, measured at a defined temperature [44,45], and normally ranges from 0.881 to 0.935 at 20 °C for vegetable oil production [46]. Our seed oils exhibited specific gravity of 0.92 (at 20 °C), and in the range of vegetable oils such as almond oil, coconut oil, cotton seed oil, grape seed oil, maize oil, sesame seed oil, soybean oil and sunflower oil [46]. Likewise, insignificantly different refractive index values, the ratio of the speed of light in a vacuum to the speed of light in the substance, among *H. heteroclita* seed oils extracted under different pretreatments were due to minimal changes in fatty acid composition related to double bonds and length of hydrocarbon chains in the triglyceride molecules [41,47]. The refractive index of our seed oils (1.47 at 40 °C) was similar to maize oil, mustard seed oil, rapeseed oil, sesame seed oil, soybean oil and sunflower seed oil (1.461–1.470 at 40 °C) [46]. Thus, the physical properties of *H. heteroclita* seed oils were similar to other vegetable oils.

For chemical properties, the moisture and volatile compounds in our seed oils (0.06–0.09% *w*/*w*) were below the maximum level specified by the Codex Alimentarius (2019) (0.20% *w*/*w*) [46], suggesting an advantage in terms of stability, with a lower risk of microbial growth and other transportation management [48,49]. Acid and peroxide values were related to the deterioration of fats and oils through hydrolysis and oxidation mechanisms, respectively [50]. Therefore, these parameters should be considered and controlled under the limits specified by the Codex Alimentarius (2019). The acid values of our seed oils were lower than 4.0 mg KOH/g, which is the maximum level of cold-pressed and virgin oils (with the exception of crude palm kernel oil and virgin palm oil) [46]. By contrast, the peroxide value of our seed oil under no pretreatment was higher than 15 mEq peroxide/kg, as the maximum level of cold-pressed and virgin oils [46], while lower peroxide values were observed in seed oils under heat-only pretreatment and the combination of heat and enzymatic pretreatments. The reduction of peroxide vales was due to the breaking down of hydroperoxides produced during heat treatment of oil to secondary oxidation products such as aldehydes, alcohols and ketones [41]. The iodine value indicates the degree of unsaturation in fatty acids. Higher iodine values increased unsaturation, with higher susceptibility to oxidation [51]. The iodine values of our seed oils (about 106 g I_2_/100 g) were in the range of almond oil, cotton seed oil, maize oil, mustard seed oil, rapeseed oil, rice bran oil and sesame seed oil (85–135 g I_2_/100 g) [46]. Saponification breaks down fats into glycerol and fatty acids by alkali treatment, with higher saponification values for oil containing high proportions of low molecular weight and short chain fatty acids [49]. The saponification values of our seed oils (about 199 mg KOH/g) were in the range of those reported in vegetable oils such as almond oil and palm oil (183–209 mg KOH/g), but lower than the saponification of coconut oil (248–265 mg KOH/g) [46]. Unsaponifiable matters are non-glyceridic substances in oil that are inert to alkaline treatment and non-volatile at 80 °C oven temperature, such as hydrocarbons, aldehydes, ketones, alcohols, sterols and pigments [52]. Pure oils mostly contain less than 2% unsaponifiable matter [53]. When compared to other vegetable oils (≤9–30 g/kg), the unsaponifiable matters of our seed oils (1.90–5.70 g/kg) were low, indicating a lower risk of contamination or adulteration [54]. The soap content is a parameter that determines the quality of oils. High soap content, especially in refined oils, represents poor quality [55]. The soap contents in our seed oils were <0.005% (*w*/*w*), which is the maximum level of soap content in vegetable oils specified by the Codex Alimentarius (2019) [46]. These results suggested that the quality of our seed oils obtained from different pretreatments, including water and volatile compounds, acid value, peroxide value and soap content were acceptable according to the specified maximum levels of vegetable oil quality characteristics by the Codex Alimentarius (2019) [46].

Proximate analysis showed that our seed oils contained 100% fat with trace amounts of other components. Vitamin E in the form of tocopherols and tocotrienols is a natural constituent found in fats and oils, with an antioxidant property that can prevent lipid peroxidation, leading to longer shelf-life and decreased low-density lipoprotein cholesterol (LDL-C) [1,56]. The vitamin E contents in our seed oils were compatible with those detected in other vegetable oils with similar fatty acid compositions, such as cottonseed oil, safflower seed oil and sunflower seed oil [57,58]. Our results also indicated that seed oil extracted using heat and enzymatic pretreatments possessed the highest δ-tocopherol. It was previously reported that δ-tocopherol exhibits higher antioxidant activities than other forms of tocopherols [1]. Therefore, our seed oil is an ideal dietary source with relatively high antioxidants.

Fatty acid profiles of our seed oils indicated 1.7-fold higher unsaturated fat than saturated fat, high polyunsaturated fat, especially omega-6 and omega-9 fatty acids, and no trans-fat. In accordance with this fatty acid profile, cottonseed oil contains high unsaturated fatty acids (70%), including 18% monounsaturated and 52% polyunsaturated fatty acids. Thus, it can be considered as naturally hydrogenated oil, with stability during frying and limited trans fatty acid formation. Moreover, its high polyunsaturated fatty acid content can prevent coronary heart disease, and cottonseed oil is considered a “Heart Oil” by the American Heart Association (AHA). Among polyunsaturated fatty acids, cottonseed oil also contains high amounts of omega-6 fatty acids with moderate amounts of omega-9 fatty acids and only trace amounts of omega-3 fatty acids [59]. Sunflower seed oil contains high amounts of polyunsaturated fatty acids (59.5%), with linoleic acid (C18:2) or omega-6 fatty acids predominant [60]. A high content of linoleic acid, along with the presence of phytosterols, plays an important role in controlling cholesterol, leading to a lower risk of heart disease [61]. Omega-6 fatty acids are essential for normal immune function and blood clotting, but excessive amounts may lead to abnormal clotting and an overactive immune system [62]. Omega-3 fatty acids can improve cardiovascular diseases, but for some types of cancers and immune systems [62], a high ratio of omega-6 to omega-3 fatty acids is associated with chronic afflictions such as cardiovascular disease, diabetes, obesity, rheumatoid arthritis and inflammatory bowel disease [63]. The recommended healthy ratio of omega-6 to omega-3 fatty acids is 1:1 to 4:1 [62]. Therefore, it is very important to balance omega-3 and omega-6 fatty acids in the diet. Omega-9 fatty acids are considered non-essential, but are a healthier choice compared to saturated fatty acids and provide several health benefits. Oils rich in oleic acid (i.e., extra virgin olive oil) can prevent inflammation, insulin resistance and oxidative stress [64].

## 5. Conclusions

Fats and oils are important ingredients in the food industry, providing beneficial nutritional and physicochemical properties related to fatty acid compositions and structures. This study developed an enzyme-assisted mechanical extraction using a mixture of Flavourzyme^®^ and Viscozyme^®^ (1:1, *w*/*w*) to enhance *H. heteroclite* seed oil extraction yield compared to non-pretreated *H. heteroclita* seed oil. The obtained oil yield was comparable to the oil yield obtained from Soxtec extraction. The optimal extraction conditions predicted by the RSM model were 2.98% enzyme loading, 48 °C incubation temperature and 76 min incubation time, giving a maximum oil extraction of 46.44%. Although enzyme assisted extraction can be expensive, costs can be compensated by an increase in extraction yield and the profit of the co-products (solid residue made of fiber and protein).

When considering the chemical properties of our heat and enzymatic pretreated seed oil, water and volatile compounds (≤0.2% *w*/*w*), acid value (≤4.0 mg KOH/g), peroxide value (≤15 mEq of active oxygen/kg) and soap content (≤0.005% *w*/*w*) were within the acceptable levels specified by the Codex Alimentarius (2019), while iodine value, saponification value and unsaponifiable matters of other vegetable oils observed by the Codex Alimentarius (2019) were in accordance with our seed oils. In terms of nutritional properties, *H. heteroclite* seed oil contained 100% fat with trace amounts of moisture, ash, carbohydrate and protein. The highest content of γ-tocopherol was found in the heat and enzymatic pretreated seed oil, suggesting a high potential for antioxidant activities. Our seed oil contained a high proportion of unsaturated to saturated fatty acids, indicating the potential to provide several health benefits against chronic diseases such as cardiovascular disease, inflammation and diabetes.

This study presents a prototype for enzyme-assisted mechanical extraction of *H. heteroclita* seed oil, to increase market value and lead to increased income for local people while creating plant sustainability.

## Figures and Tables

**Figure 1 foods-12-00292-f001:**
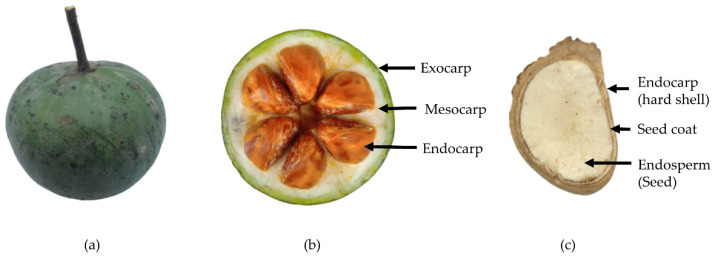
The figure illustrates (**a**) whole fruit, (**b**) sectioned fruit and (**c**) sectioned seed of *H. heteroclita*.

**Figure 2 foods-12-00292-f002:**
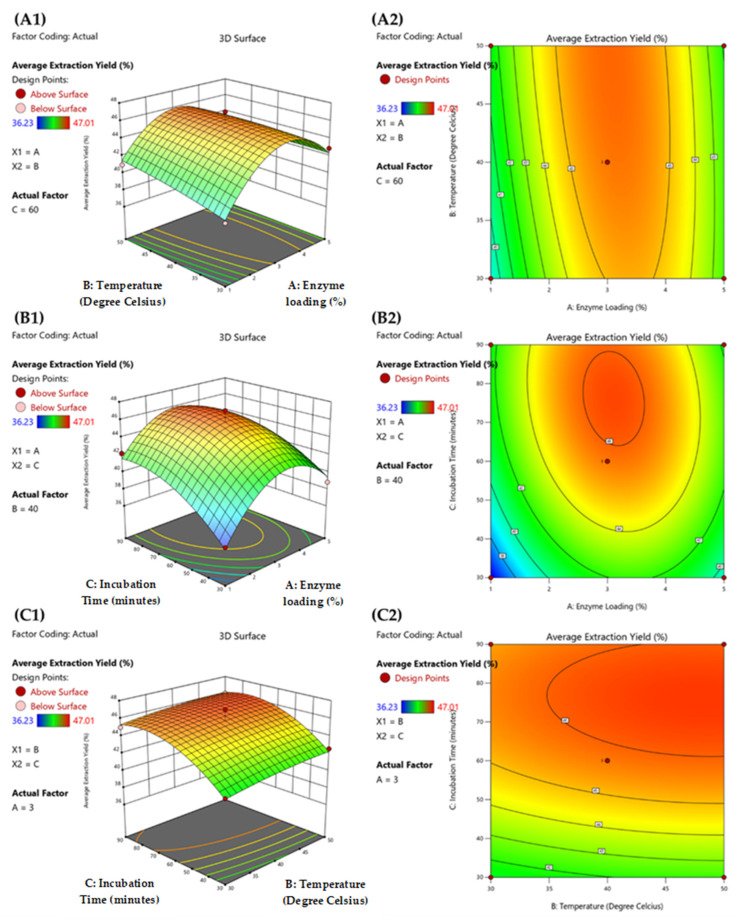
(**A1**,**A2**) Three-dimensional surface plot and contour plot of enzyme loading (%) and incubation temperature (°C) with 60 min of incubation time, (**B1**,**B2**) enzyme loading (%) and incubation time (h) at a constant incubation temperature (40 °C) and (**C1**,**C2**) incubation temperature (°C) and incubation time (h) with 3% (*w*/*w*) enzyme loading.

**Table 1 foods-12-00292-t001:** Influence of various enzymes on *H. heteroclita* seed oil extraction.

Extraction	Pretreatments	Oil Yield (%)	Extraction Efficiency (%)
Mechanical extraction	Heat + Flavourzyme^®^	38.10 ± 1.03 ^ab^	97.29
	Heat + Viscozyme^®^	36.23 ± 1.10 ^b^	92.52
	Heat + Flavourzyme^®^ + Viscozyme^®^ (1:1, *w*/*w*)	38.94 ± 0.93 ^a^	99.44
	None	33.64 ± 1.90 ^c^	85.90
Soxtec extraction	None	39.16 ± 0.27 ^a^	100.00

All data are shown as the mean ± standard deviation (SD) of triplicate determinations (*n* = 3). Different lowercase letters denote significantly different oil yield (%) at *p* < 0.05 of the different oil extraction methods using one-way ANOVA, followed by Duncan’s multiple comparison test.

**Table 2 foods-12-00292-t002:** Box–Behnken design matrix in coded and uncoded values and response.

Run	Coded Variables			Uncoded Variables			Response
	X_1_	X_2_	X_3_	Enzyme Loading (%)	Incubating Temperature (°C)	Incubating Time (h)	Oil Yield (%)
1	−1	−1	0	1	30	1.0	38.93
2	+1	−1	0	5	30	1.0	42.88
3	−1	+1	0	1	50	1.0	40.96
4	+1	+1	0	5	50	1.0	42.20
5	−1	0	−1	1	40	0.5	36.23
6	+1	0	−1	5	40	0.5	38.83
7	−1	0	+1	1	40	1.5	42.18
8	+1	0	+1	5	40	1.5	41.56
9	0	−1	−1	3	30	0.5	41.29
10	0	+1	−1	3	50	0.5	42.56
11	0	−1	+1	3	30	1.5	44.99
12	0	+1	+1	3	50	1.5	45.94
13	0	0	0	3	40	1.0	44.90
14	0	0	0	3	40	1.0	45.29
15	0	0	0	3	40	1.0	47.01

X_1_: enzyme loading (%); X_2_: incubation temperature (°C); X_3_: incubation time (h).

**Table 3 foods-12-00292-t003:** ANOVA for RSM model.

Source	SS	df	MS	*F*-Value	*p*-Value
Model	118.08	9	13.12	15.53	0.0038
X_1_	6.43	1	6.43	7.61	0.0399
X_2_	1.59	1	1.59	1.89	0.2280
X_3_	31.05	1	31.05	36.76	0.0018
X_1_X_2_	1.84	1	1.84	2.17	0.2004
X_1_X_3_	2.59	1	2.59	3.07	0.1402
X_2_X_3_	0.0256	1	0.0256	0.0303	0.8686
X_1_^2^	66.47	1	66.47	78.69	0.0003
X_2_^2^	0.2269	1	0.2269	0.2687	0.6263
X_3_^2^	11.84	1	11.84	14.01	0.0134
Residual	4.22	5	0.8447		
Lack of fit	1.70	3	0.5675	0.4503	0.7440
Pure error	2.52	2	1.26		
Cor total	122.30	14			

X_1_: enzyme loading (%); X_2_: incubation temperature (°C); X_3_: incubation time (h); SS: sum of squares; df: degree of freedom; MS: mean square. Standard deviation (SD): 0.92; mean: 42.38; coefficient of variation %: 2.17; coefficient of determination (R^2^): 0.96; adjusted R^2^: 0.90; predicted R^2^: 0.73; adequate precision: 13.33.

**Table 4 foods-12-00292-t004:** Physical properties of *H. heteroclita* seed oils under different mechanical extraction pretreatment conditions.

Physical Properties		No Pretreatment	Heat Pretreatment ^$^	Heat and Enzymatic Pretreatments ^#^
Color	L*	65.10 ± 0.29 ^ab^	64.96 ± 0.24 ^b^	65.67 ± 0.39 ^a^
	a*	−3.34 ± 0.04 ^b^	−3.09 ± 0.07 ^ab^	−2.07 ± 0.96 ^a^
	b*	18.54 ± 1.95 ^ab^	25.51 ± 2.80 ^a^	16.70 ± 5.38 ^b^
	YI	40.69 ± 4.45 ^ab^	56.11 ± 6.36 ^a^	36.40 ± 11.99 ^b^
Viscosity (cps)		62.71 ± 2.28 ^b^	54.91 ± 0.43 ^c^	88.42 ± 1.00 ^a^
Specific gravity (at 20 °C) ^ns^		0.92 ± 0.00	0.92 ± 0.00	0.92 ± 0.00
Refractive index (at 40 °C) ^ns^		1.47 ± 0.00	1.47 ± 0.00	1.47 ± 0.00

All data are shown as the mean ± standard deviation (SD) of triplicate determinations (*n* = 3). Different lowercase letters denote significantly different values of the same physical property at *p* < 0.05, while ^ns^ denotes no significantly different values of the same physical property at *p* ≥ 0.05 of seed oils which underwent different extraction conditions using one-way ANOVA, followed by Duncan’s multiple comparison test; ^$^ denotes seed oil obtained from drying at 55 °C until reaching 10% moisture content; ^#^ denotes seed oil obtained from heat and enzymatic pretreatments (optimal enzymatic extraction condition: 2.98% (*w*/*w*) enzyme loading, 48 °C of incubation temperature and 76 min of incubation time). Color was expressed in CIELAB units, where L* represents dark (0) to light (100), a* represents green (−) to red (+) colors, and b* represents blue (−) to yellow (+) colors; YI: yellowness index; cps: centipoise.

**Table 5 foods-12-00292-t005:** Chemical properties of *H. heteroclita* seed oils under different mechanical extraction pretreatment conditions.

Chemical Properties	No Pretreatment	Heat Pretreatment ^$^	Heat and Enzymatic Pretreatments ^#^
Water and volatile compounds (% *w*/*w*) ^ns^	0.09 ± 0.01	0.06 ± 0.01	0.06 ± 0.01
Acid (mg KOH/g)	0.96 ± 0.08 ^a^	0.46 ± 0.02 ^b^	0.26 ± 0.02 ^c^
Peroxide (mEq peroxide/kg)	22.32 ± 0.15 ^a^	3.85 ± 0.03 ^c^	5.29 ± 0.23 ^b^
Iodine (g I_2_/100 g) ^ns^	105.75 ± 0.35	106.40 ± 0.14	105.75 ± 0.35
Saponification (mg KOH/g)	199.20 ± 0.28 ^a^	199.10 ± 0.28 ^a^	198.00 ± 0.00 ^b^
Unsaponifiable matters (g/kg)	1.90 ± 0.02 ^c^	2.64 ± 0.01 ^b^	5.70 ± 0.01 ^a^
Soap content (% *w*/*w*)	0.0025 ± 0.00 ^a^	0.0035 ± 0.00 ^a^	0.0004 ± 0.00 ^b^

All data are shown as the mean ± standard deviation (SD) of triplicate determinations (*n* = 3). Different lowercase letters denote significantly different contents of the same chemical property at *p* < 0.05, while ^ns^ denotes no significantly different contents of the same chemical property at *p* ≥ 0.05 of seed oils which underwent different mechanical extraction conditions using one-way ANOVA, followed by Duncan’s multiple comparison test. ^$^ denotes seed oil obtained from drying at 55 °C until reaching 10% moisture content; ^#^ denotes seed oil obtained from heat and enzymatic pretreatments (optimal enzymatic extraction condition: 2.98% (*w*/*w*) enzyme loading, 48 °C of incubation temperature and 76 min of incubation time). KOH: potassium hydroxide; mEq: milliequivalents.

**Table 6 foods-12-00292-t006:** Nutritional properties of *H. heteroclita* seed oils under different extraction pretreatment conditions.

Nutritional Properties		No Pretreatment	Heat Pretreatment ^$^	Heat and Enzymatic Pretreatments ^#^
Moisture content (g/100 g)		<0.01	<0.01	<0.01
Ash (g/100 g)		<0.01	<0.01	<0.01
Carbohydrate (g/100 g)		<0.01	<0.01	<0.01
Protein (g/100 g)		<0.10	<0.10	<0.10
Fat (g/100 g) ^ns^		100.00 ± 0.00	100.00 ± 0.00	100.00 ± 0.00
Vitamin E (mg/100 g)	α-tocopherol	0.03 ± 0.00	nd	nd
	β-tocopherol	0.12 ± 0.01	0.03 ± 0.00	nd
	γ-tocopherol	nd	0.08 ± 0.00	3.52 ± 0.38
	δ-tocopherol	0.18 ± 0.01 ^c^	1.58 ± 0.08 ^b^	88.29 ± 0.37 ^a^
	α-tocotrienol	nd	nd	3.56 ± 0.58
	β-tocotrienol	0.01 ± 0.00	nd	0.65 ± 0.03
	γ-tocotrienol	nd	nd	2.50 ± 0.08
	δ-tocotrienol	0.88 ± 0.01	9.84 ± 0.33	nd

All data are shown as the mean ± standard deviation (SD) of triplicate determinations (*n* = 3). Different lowercase letters denote significantly different contents of the same nutritional composition at *p* < 0.05, while ^ns^ denotes no significantly different contents of the same nutritional composition at *p* ≥ 0.05 of seed oils which underwent different mechanical extraction conditions using one-way ANOVA, followed by Duncan’s multiple comparison test. ^$^ denotes seed oil obtained from drying at 55 °C until reaching 10% moisture content; ^#^ denotes seed oil obtained from heat and enzymatic pretreatments (optimal enzymatic extraction condition: 2.98% (*w*/*w*) enzyme loading, 48 °C of incubation temperature and 76 min of incubation time). nd: not detected.

**Table 7 foods-12-00292-t007:** Fatty acid composition (g/100 g) and omega-3,6,9 fatty acids (mg/100 g) of *H. heteroclita* seed oils under different mechanical extraction pretreatment conditions.

Fatty Acid Profile	No Pretreatment	Heat Pretreatment ^$^	Heat and Enzymatic Pretreatments ^#^
**Saturated fat ^ns^**	**35.67 ± 0.51**	**34.79 ± 0.28**	**34.32 ± 0.33**
Butyric acid (C4:0)	nd	nd	nd
Caproic acid (C6:0)	nd	nd	nd
Caprylic acid (C8:0)	0.02 ± 0.00	nd	0.02 ± 0.00
Capric acid (C10:0)	nd	nd	nd
Undecanoic acid (C11:0)	nd	nd	nd
Lauric acid (C12:0)	0.02 ± 0.02	nd	nd
Tridecanoic acid (C13:0)	nd	nd	nd
Myristic acid (C14:0) ^ns^	0.07 ± 0.01	0.08 ± 0.01	0.07 ± 0.00
Pentadecanoic acid (C15:0)	nd	nd	0.01 ± 0.00
Palmitic acid (C16:0) ^ns^	27.24 ± 0.35	26.00 ± 0.10	26.34 ± 0.66
Heptadecanoic acid (C17:0) ^ns^	0.08 ± 0.00	0.08 ± 0.00	0.08 ± 0.01
Stearic acid (C18:0)	6.85 ± 0.06 ^b^	7.77 ± 0.28 ^a^	6.80 ± 0.04 ^b^
Arachidic acid (C20:0) ^ns^	0.46 ± 0.01	0.48 ± 0.01	0.44 ± 0.01
Heneicosanoic acid (C21:0)	nd	nd	nd
Behenic acid (C22:0) ^ns^	0.26 ± 0.02	0.22 ± 0.01	0.23 ± 0.00
Tricosanoic acid (C23:0)	nd	nd	nd
Lignoceric acid (C24:0)	0.09 ± 0.00 ^a^	0.07 ± 0.00 ^b^	0.08 ± 0.01 ^b^
**Monounsaturated fatty acid**	**11.85 ± 0.11 ^b^**	**12.63 ± 0.17 ^a^**	**11.71 ± 0.27 ^b^**
Myristoleic acid (C14:1)	nd	nd	nd
Pentadecenoic acid (C15:1)	nd	nd	nd
Palmitoleic acid (C16:1n7) ^ns^	0.12 ± 0.01	0.12 ± 0.00	0.11 ± 0.00
Heptadecenoic acid (C17:1)	nd	nd	nd
Elaidic acid (C18:1n9t)	nd	nd	nd
Oleic acid (C18:1n9c) (omega-9) ^ns^	11.28 ± 0.44	12.46 ± 0.19	11.46 ± 0.18
Eicosenoic acid (C20:1n9) (omega-9) ^ns^	0.08 ± 0.01	0.08 ± 0.01	0.08 ± 0.00
Erucic acid (C22:1n9) (omega-9)	nd	nd	nd
Nervonic acid (C24:1n9) (omega-9)	nd	nd	nd
**Polyunsaturated fatty acid**	**48.22 ± 0.22 ^b^**	**48.02 ± 0.28 ^b^**	**49.29 ± 0.34 ^a^**
Linolelaidic acid (C18:2n6t)	nd	nd	nd
Linoleic acid (C18:2n6c) (omega-6) ^ns^	47.84 ± 0.22	47.39 ± 0.52	48.19 ± 1.13
γ-Linolenic acid (C18:3n6) (omega-6)	nd	nd	nd
α-Linolenic acid (C18:3n3) (ALA, omega-3)	0.06 ± 0.00	0.06 ± 0.00	0.06 ± 0.00
Eicosadienoic acid (C20:2n6) (omega-6)	nd	nd	nd
Eicosatrienoic acid (C20:3n6) (omega-6)	nd	nd	nd
Eicosatrienoic acid (C20:3n3) (omega-3)	nd	nd	nd
Arachidonic acid (C20:4n6) (ARA, omega-6)	nd	nd	nd
Docosadienoic acid (C22:2n6) (omega-6)	nd	nd	nd
Eicosapentaenoic acid (C20:5n3) (EPA, omega-3)	nd	nd	nd
Docosahexaenoic acid (C22:6n3) (DHA, omega-3)	nd	nd	nd
**Unsaturated fat ^ns^**	**59.93 ± 0.14**	**60.27 ± 0.42**	**60.38 ± 0.27**
**Trans fat**	**nd**	**nd**	**nd**
Omega-3	57.70 ± 0.04 ^a^	56.18 ± 0.35 ^b^	58.15 ± 0.27 ^a^
Omega-6 ^ns^	47,846.20 ± 215.26	47,388.29 ± 521.02	48,187.86 ± 1128.16
Omega-9	11,362.36 ± 427.61 ^b^	12,536.92 ± 127.36 ^a^	11,533.04 ± 179.92 ^b^

All data are shown as the mean ± standard deviation (SD) of triplicate determinations (*n* = 3). Different lowercase letters denote significantly different contents of the same fatty acid at *p* < 0.05, while ^ns^ denotes no significantly different contents of the same fatty acid at *p* ≥ 0.05 among seed oils which underwent different mechanical extraction conditions using one-way ANOVA, followed by Duncan’s multiple comparison test. ^$^ denotes seed oil obtained from drying at 55 °C until reaching 10% moisture content; ^#^ denotes seed oil obtained from heat and enzymatic pretreatments (optimal enzymatic extraction condition: 2.98% (*w*/*w*) enzyme loading, 48 °C of incubation temperature and 76 min of incubation time); nd: not detected, with limit of detection (LOD) at 0.01 g/100 g.

## Data Availability

Data are contained within this article.

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
