# Peer review of "Optimization of Enzyme-Assisted Mechanical Extraction Process of Hodgsonia heteroclita Oilseeds and Physical, Chemical, and Nutritional Properties of the Oils"

_foods, 2023, doi:10.3390/foods12020292_

Round 1

Reviewer 1 Report

The idea to study the oil extraction from neglected and underutilized crops is an interesting one and it is part of the concern for sustainable development. The description of the methods is clear, and the results obtained are correctly interpreted.

The paper could be published after minor revision

1)       Commonly, oil extraction methods use an organic solvents. /Line 58

2)       Equation 5 must be written after ANOVA analysis in the form in which only the significant terms will appear, for which (p<0.005)

3)       In Figure 2 for the image A1, B1 and C1 the writing is not clear. For example, incubation time, enzyme loading are difficult to read.

4)       I agree with enzyme assisted extraction, but the enzymes are not cheap substances and perhaps it is necessary to add some words about the economical cost of this technique if the authors desire to apply it at industrial scale.

Author Response

Best regards,

Nattira

Reviewer 2 Report

The article with the title "Optimization of Enzyme-Assisted Mechanical Extraction Process of Hodgsonia heteroclita Oilseeds and Physical, Chemical, and Nutritional Properties of The Oils" has 23 pages, 7 Tables, 2 Figures and 89 References. 

Unfortunately, the article sent to the editorial office of this journal is not written carefully and does not meet the requirements for accept.

The article with the title "Optimization of Enzyme-Assisted Mechanical Extraction Process of Hodgsonia heteroclita Oilseeds and Physical, Chemical, and Nutritional Properties of The Oils" has 23 pages, 7 Tables, 2 Figures and 89 References. 

Unfortunately, the article sent to the editorial office of this journal is not written carefully and does not meet the requirements for accept.

The number of 23 pages is unnecessary, due to the extent of the width of the experiment, that has been made. The detriment of the article is to establish an attempt and to evaluate commercial enzymes. The conclusion and the scope of the experiment is like "Product Development study".

The Introduction chapter has more than 5,000 characters /including spaces/ and 17 references.

In the Material and Methods chapter, the authors referred to a 15 years old study in the case of preparation of material. (De Wilde, W.J.O.; Duyfjes, B.E.e. cucurbitaceae. Flora of Thailand 2008, 9, 411-546).

The measurement and methodology of the experiment has considerable shortcomings. The number of measurements and samples, expression of repetitions, etc. is missing.

There is no description of the measurement and reference to use the Colorflex® EZ Spectrophotometer and the necessary description for colour measurement. Is the device used for this type of experiments? This is complication, if the another scientist could do this experiment and compare data.

On the other hand, Chapter 2.5 could be less detailed and with more reference. /I'm not sure there are some passages from other publications./

The discussion is very detailed and long, with more than 40 references, which is unnecessary even for very fundamental studies.

The authors do not use the MDPI Journals reference style /in template/.

Therefore, I recommend reject the article to hold a high standard of articles of this Journal.

Author Response

Best regards,

Nattira

Reviewer 3 Report

This is a well-written manuscript on the enzyme-assisted mechanical extraction process of Hodgsonia heteroclita Oilseeds. After carefully reading the documents, I have a few questions that could help improve the quality of the manuscript.

1. Hodgsonia heteroclita is a species that grows in a specific area of ​​Thailand. To increase the interest and application of the document, I think it should be indicated in the Introduction that it belongs to the Cucurbitaceae family, in which we can find numerous species of economic interest worldwide.

2. The optimal pH range for both enzymes is pH 3.5–7, but what is the pH of the sample before pretreatment?

3. The references are excessive and should be reduced. For example:

  • Line 36, references 2 and 3 are cited. Only reference 3 should go, which cites the report from IndustryARCTM. I
  • Line 65, references 6-8 could be replaced by a review such as Processes 2021, 9, 1839.

4. References must be homogenized. According to the instructions for authors, the name of the journal should be abbreviated.

Author 1, A.B.; Author 2, C.D. Title of the article. Abbreviated Journal Name YearVolume, page range.

Author Response

Best regards,

Nattira

Round 2

Reviewer 2 Report

The article was revised in terms of remarks in review. I don't have any comments.

Author Response

Dear Reviewer,

Thank you for consideration of our revised manuscript.

Sincerely,

Nattira